# Defining cardiac cell populations and relative cellular composition of the early fetal human heart

**Jennifer M. Dewing**[1]*, **Vinay Saunders**[1], **Ita O'Kelly**[1,2], **David I. Wilson**[1]

**1** Institute for Developmental Sciences, School of Human Development and Health, Faculty of Medicine, University of Southampton, Southampton, United Kingdom, **2** Immunocore Ltd, Abingdon, Oxford, United Kingdom

* jmd2g08@soton.ac.uk

**Data Availability Statement:** All relevant data are within the manuscript and its Supporting Information files.

## Abstract

While the adult human heart is primarily composed of cardiomyocytes, fibroblasts, endothelial and smooth muscle cells, the cellular composition during early development remains largely unknown. Reliable identification of fetal cardiac cell types using protein markers is critical to understand cardiac development and delineate the cellular composition of the developing human heart. This is the first study to use immunohistochemistry (IHC), flow cytometry and RT-PCR analyses to investigate the expression and specificity of commonly used cardiac cell markers in the early human fetal heart (8–12 post-conception weeks). The expression of previously reported protein markers for the detection of cardiomyocytes (Myosin Heavy Chain (MHC) and cardiac troponin I (cTnI), fibroblasts (DDR2, THY1, Vimentin), endothelial cells (CD31) and smooth muscle cells (α-SMA) were assessed. Two distinct populations of cTnI positive cells were identified through flow cytometry, with MHC positive cardiomyocytes showing high cTnI expression (cTnI$^{High}$) while MHC negative non-myocytes showed lower cTnI expression (cTnI$^{Low}$). cTnI expression in non-myocytes was further confirmed by IHC and RT-PCR analyses, suggesting troponins are not cardiomyocyte-specific and may play distinct roles in non-muscle cells during early development. Vimentin (VIM) was expressed in cultured ventricular fibroblast populations and flow cytometry revealed VIM$^{High}$ and VIM$^{Low}$ cell populations in the fetal heart. MHC positive cardiomyocytes were VIM$^{Low}$ whilst CD31 positive endothelial cells were VIM$^{High}$. Using markers investigated within this study, we characterised fetal human cardiac populations and estimate that 75–80% of fetal cardiac cells are cardiomyocytes and are MHC$^+$/cTnI$^{High}$/VIM$^{Low}$, whilst non-myocytes comprise 20–25% of total cells and are MHC$^-$/cTnI$^{Low}$/VIM$^{High}$, with CD31$^+$ endothelial cells comprising ~9% of this population. These findings show distinct differences from those reported for adult heart.

## Introduction

The cellular composition of the adult mammalian heart has been defined through years of study and is known to be primarily composed of cardiomyocytes, fibroblasts, endothelial and

**Funding:** The study was funded by the Medical Research Council (MRC), through Medical Research Council Capacity Building Studentship (Grant No. G1000406) received by the University of Southampton and was used to support the PhD studentship of JD and by A Joint MRC/Wellcome Trust Grant (Grant No. 099175/Z/12/Z) received by Newcastle University that was used to fund the HDBR tissue for this project. Immunocore also provided funding in the form of salary for IO. The specific roles of these authors are articulated in the 'author contributions' section. The funders had no role in study design, data collection and analysis, decision to publish, or preparation of the manuscript.

**Competing interests:** The authors have read the journal's policy and have the following competing interests: IO is a paid employee of Immunocore. There are no patents, products in development or marketed products associated with this research to declare. This does not alter our adherence to PLOS ONE policies on sharing data and materials.

smooth muscle cells [1–7]. Rodent studies have proposed that cardiomyocytes occupy ~75% of normal adult myocardial volume but account for only 30–40% of cell number, with non-cardiomyocytes (endothelial, smooth muscle cells and fibroblasts), though smaller in size, being the predominant cell type. Human studies have similarly concluded that the adult heart consists of approximately 30% cardiomyocytes and 50% endothelial cells [4]. While we have a good understanding of adult cardiac cellular composition, less is known about fetal heart composition, which is likely to differ due to the dynamic stages of early development. Delineating the cellular composition of the human fetal heart will influence our ability to understand normal human cardiac development. Furthermore, given that pluripotent stem cell-derived cardiomyocytes (PSC-CMs) have been shown to be phenotypically similar to cardiomyocytes of the mid-gestation human fetal heart, an improved understanding of the protein marker profiles of these cells could be critical to further develop and validate the cellular phenotype of differentiated PSC-CMs and provide additional tools for their purification and use in regenerative medicine [8].

The identification, characterisation and purification of cardiac cells relies upon their expression of cell-specific protein markers. Commonly used markers of cardiomyocytes include the sarcomeric proteins cardiac troponin (subunits I, T and C), myosin and tropomyosin. α-myosin heavy chain (α-MHC) has high expression levels in cardiac muscle and significantly lower levels in skeletal muscle; consequently the α-MHC promoter is often used in transgenic mouse models to track cardiomyocytes [9–12]. Endothelial and smooth muscle cells that line the cardiac vasculature can be detected by their expression of CD31 (PECAM-1) and α-smooth muscle actin (α-SMA), respectively. However, CD31 is also present on macrophages, and α-SMA is upregulated in activated fibroblasts, termed myofibroblasts, that respond to injured cardiac tissue. Whilst these markers may identify mature, differentiated endothelial, smooth muscle and cardiomyocyte cell populations in adult tissue, their specificity during development may be distinct.

One of the key challenges to the isolation and identification of cell populations in the heart is the heterogeneity of fibroblast cells and the lack of a defined cardiac fibroblast-specific marker. Fibroblasts are cells of mesenchymal origin that produce extracellular matrix proteins including collagen and fibronectin [13]. The phenotypic plasticity of cardiac fibroblasts depends on their developmental origin (epicardium, endocardium or neural crest) as well as their location within the myocardium [14]. The cell surface collagen receptor DDR2 has previously been used as a marker of cardiac fibroblasts [15–18]. However, although previous studies have shown DDR2 to be absent from cardiomyocytes, its expression has also been reported in rat endothelial and smooth muscle cells, raising the possibility of interspecies differences [2, 17, 19, 20]. The cytoskeletal intermediate filament vimentin is a commonly used fibroblast marker due to its high levels of expression in cells of mesenchymal origin. However, vimentin may not provide the level of specificity previously attributed to this marker due to its documented expression in endothelial and smooth muscle cells [21, 22]. Thy-1 (CD90), a cell-surface glycoprotein, has been detected in cultured rat cardiac fibroblasts, with increased expression detected in fibrotic areas within the heart, suggesting it may be a marker for proliferating fibroblasts [9, 23, 24]. Nonetheless, Thy-1 has also been detected in thymocytes, T-cells, neurons, hematopoietic stem cells and endothelial cells [25–27]. Whilst Pinto *et* al., were able to use Thy-1 and Sca-1 to delineate cardiac fibroblasts in mice, neither of these markers were successful at isolating cardiac fibroblasts of the adult human heart [4]. Similarly, fibroblast-specific protein 1 (FSP1) has been shown to lack fibroblast specificity in cardiac tissue during remodelling, with expression identified in hematopoietic, endothelial and vascular smooth muscle cells [28]. Together, these studies suggest that many commonly used markers of fibroblasts lack specificity.

The aim of this study was to use immunohistochemistry (IHC) and flow cytometry, alongside RT-PCR analysis, to estimate the cellular composition of the early fetal human heart and, in doing so, evaluate the specificity of a range of markers that hold the potential to define novel marker profiles of cardiac cell populations during early development.

## Materials and methods

### Isolation of fetal human cardiac cells

Human fetal heart tissue (8 to 12 post conception weeks (pcw) was obtained from the Human Developmental Biology Resource (HDBR), Newcastle, UK, following informed written consent. Tissue collection was in agreement with the Declaration of Helsinki (ethics approval reference: 08/HO906/21+5 NRES Committee North East- Newcastle & North Tyneside). The fetal tissues comprised the atria, ventricles, pulmonary artery and arch of aorta; the tissues used for immunohistochemistry were fixed without dissection and embedded in paraffin (see Fig 1 and immunohistochemistry methods below). For flow cytometry, the aorta and pulmonary artery were removed by careful dissection and processed separately from the heart, comprising the ventricles and atria. In order to remove blood cells contained within the chambers prior to flow cytometry, the atria were dissected from the ventricles to facilitate rinsing of the tissues. Ventricles and atria were divided into 1mm$^3$ pieces using a tissue chopper (McIlwain), combined, and mechanically dissociated together in 3ml of PBS using the gentleMACS dissociator machine (Miltenyl Biotec) using the programme pre-set for heart tissue. The cell suspension (combined ventricle-atria cells) was passed through a 70 μm nylon cell strainer (BD Biosciences). For analysis of cells from the aorta and pulmonary artery, the process was

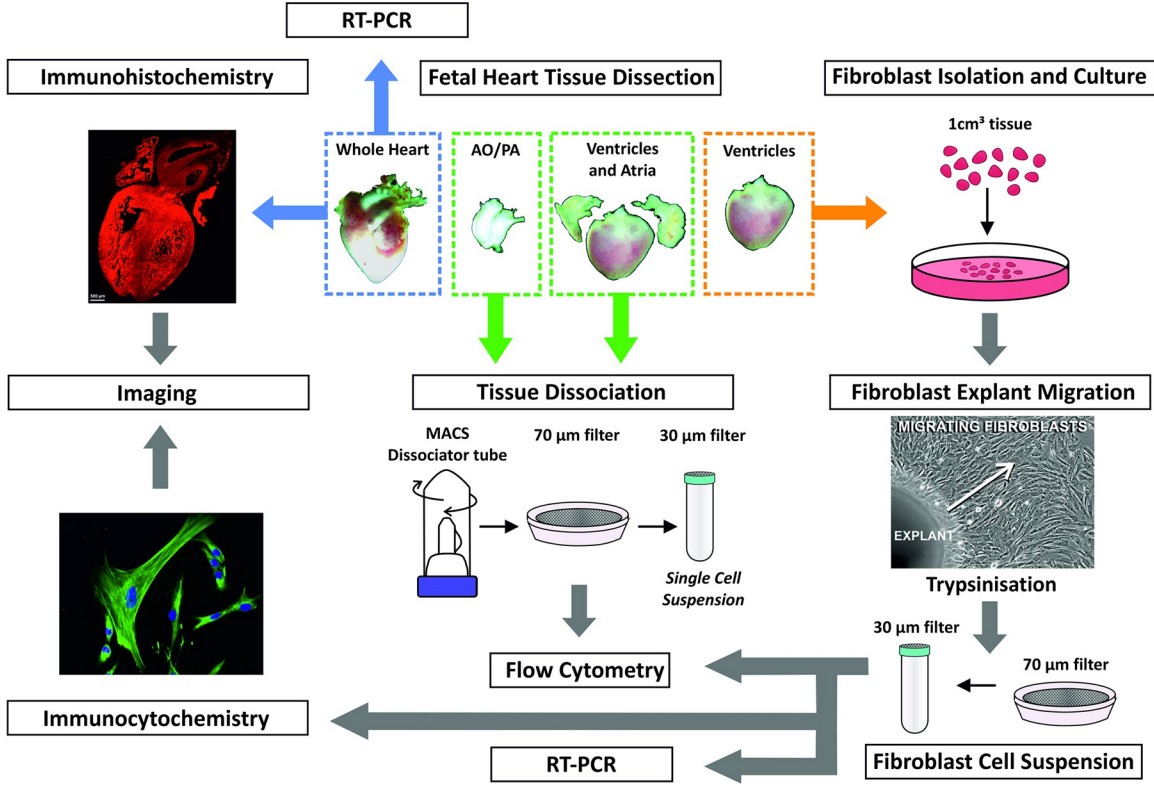

**Fig 1.**

repeated with dissected aorta and pulmonary artery tissue, which enabled the investigation of cells uncontaminated by myocardial content.

## Antibodies for flow cytometry and immunohistochemistry

Antibodies were selected to identify cardiomyocytes, fibroblasts, smooth muscle cells and endothelial cells, based on current literature [12, 16, 17, 23, 29]. See S1 Table for complete list of primary and secondary antibodies used and their corresponding controls.

## Flow cytometry

The cardiac cell suspensions were prepared for flow cytometry by filtering through a 35 μm cap before washing in 1% fetal calf serum in PBS. Centrifugation of cells between washes was carried out at 300 g. Cells were treated with human FcR blocking serum (BD Bisociences) for 15 minutes on ice, washed and fixed in 1% formaldehyde (Sigma) for 10 minutes at room temperature. Cells were washed and permeabilized in 0.05% saponin (Sigma) for 20 minutes at room temperature. For cell surface staining, permeabilisation was omitted. Cells were incubated with primary antibodies on ice for 1hr, then washed three times. For non-conjugated antibodies, cells were subsequently incubated with secondary antibodies for 30 minutes on ice, followed by further washing. Cells were analysed using a BD FACSCanto I Flow cytometer (BD biosciences). Single and dual stain flow cytometry analyses were carried out to target the major cardiac cells types. Small size exclusion was performed on forward scatter and side scatter dot plots to select the cell population (S1A Fig). Representative flow cytometry histograms showing the fluorescence intensity of unstained cardiac cells is displayed in S1B–S1D Fig. Isotype antibodies were used as controls for directly conjugated primary antibodies. Incubation with secondary antibodies only was used as a control for samples treated with primary and secondary antibodies. Gating of positive cells was determined using flow cytometry dot plots of negative control samples verses stained samples: positive cells were selected based on their expression above the negative control. This gating strategy is displayed in S2 Fig. For dual flow cytometry, compensation analysis was applied to samples based on unstained and single stained controls.

## Immunohistochemistry

Immunohistochemistry followed previously reported methods [30]. Briefly, fetal human heart tissue was fixed in 4% formaldehyde, embedded in paraffin and 10 μm thick microtome sections cut. Antigen retrieval was performed by boiling sections in 0.1mM sodium citrate solution (pH 9.0) + 0.05% Tween 20 for 15 minutes. Sections were then stained with primary antibodies and incubated overnight at 4°C in a moist chamber. For non-conjugated antibodies, sections were washed and incubated with secondary antibodies for 2 hours at room temperature. Nuclei were stained by incubation with 4', 6-diamidino-2-pheylindole (DAPI), followed by mounting. Representative images of negative control staining are shown in S3 Fig.

## Cardiac fibroblast Isolation and cell culture

Fibroblasts were isolated from fetal human heart tissue by explant migration following previously reported methods by Ieda *et al.*, (2010) [9]. Briefly, following dissection, ventricles were chopped into 1mm$^3$ pieces using a tissue chopper (McIlwain) and cultured in 10% FBS DMEM on gelatin-coated plates. Following the migration of fibroblasts from the explant (2 weeks), the cells and explants were trypsinised from the plate, filtered through a 70 μm nylon cell strainer to exclude explant tissue and re-seeded on gelatin-coated 6-well tissue culture

plates containing 10% FBS DMEM until 80% confluent. Cells were pelleted for RNA exaction and flow cytometry or re-seeded on glass coverslips for immunocytochemistry.

## Immunocytochemistry

Cardiac fetal fibroblasts cultured on glass coverslips in 6-well plates were washed with PBS and fixed with 4% formaldehyde for 7 minutes. The cells were washed thoroughly with PBS before quenching the PFA with 100 mM glycine for 10 minutes. Cells were permeabilized with 0.1% triton-X-100 for 7 minutes followed by washing with PBS and blocking with 3% BSA diluted in PBS for 1 hour. Primary antibody diluted in 0.1% triton containing 3% goat serum was added to wells before being incubated in a humidity chamber overnight at 4˚C. Cells were thoroughly washed with PBS and diluted secondary antibody added to each well and incubated at room temperature in the dark for two hours. The cells were washed with PBS three times before mounting with vecta-shield with DAPI.

## Reverse Transcriptase (RT)-PCR

Total RNA was isolated using TRIzol Reagent (Invitrogen) from whole heart tissue and cell pellets of cultured primary ventricular fibroblasts. 1 µg of RNA was used for cDNA synthesis using M-MLV reverse transcriptase (Promega). RT-PCR was carried out using Go-Taq polymerase (Promega) in a 50 µl reaction containing 0.5 µM primers, 10 mM dNTPs and 5 µl of 10 ng/µl cDNA. Primer sequences and amplicon sizes are listed in S2 Table. Polymerase Chain Reaction (PCR) consisted of initial denaturation of DNA at 95˚C, followed by 40 cycles of 94˚C for 1 minute, annealing at 58˚C for 1 minute and extension at 72˚C for 1 minute, with a final extension of 72˚C for 10 minutes. The PCR products were resolved on 1.5% agarose gels containing Nancy-520 (Sigma Aldrich) and imaged using a high performance ultraviolet trans-illuminator (UVP) and the associated DocIT software.

## Results

### Single marker immunohistochemistry and flow cytometry of the fetal human heart reveals cell populations expressing high and low cTnI levels

We sought to define fetal cardiac cell populations using immunohistochemistry and flow cytometry. The experimental strategy used to investigate the different populations in the early developing heart is summarised in Fig 1. Initially, immunohistochemistry of whole fetal human heart sections were used to investigate the expression of commonly used markers of the four main cardiac cell types: cardiomyocytes (MHC (MF20), cTnI (TNNI3)), cardiac fibroblasts (vimentin (VIM), DDR2, THY1/CD90), endothelial cells (CD31/PECAM-1)) and smooth muscle cells (α-SMA) (Fig 2). cTnI and MHC showed distinct sarcomeric staining typical of cardiomyocytes (Fig 2A). The fibroblast markers showed extensive staining throughout the heart tissue, with vimentin localised to the cytoplasm and the membrane protein THY1 also showing cytoplasmic staining (Fig 2B). DDR2 expression was seen throughout the myocardium and was localised to the nucleus and the cytoplasm. CD31 and α-SMA showed positive staining of the endothelial and smooth muscle cells lining the vessels in the heart, respectively (Fig 2C).

To determine the relative protein expression levels of these cardiac markers at the cellular level, flow cytometry was performed on ventricle-atria cell suspensions of fetal hearts. Flow cytometry dot plots of negative controls (isotype controls and secondary antibody only) are displayed in S2 Fig. Our strategy to determine the proportions of specific cardiac cell types within the fetal human heart was based on the percentage of cells exhibiting expression above the negative control, which can be visualised by the flow cytometry histograms presented in Fig 3. Using

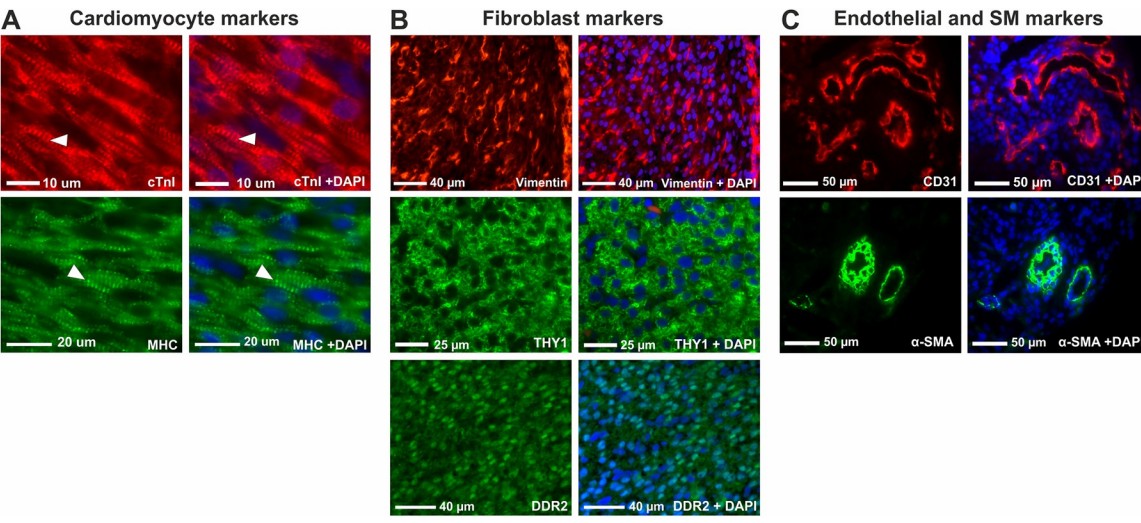

**Fig 2.**

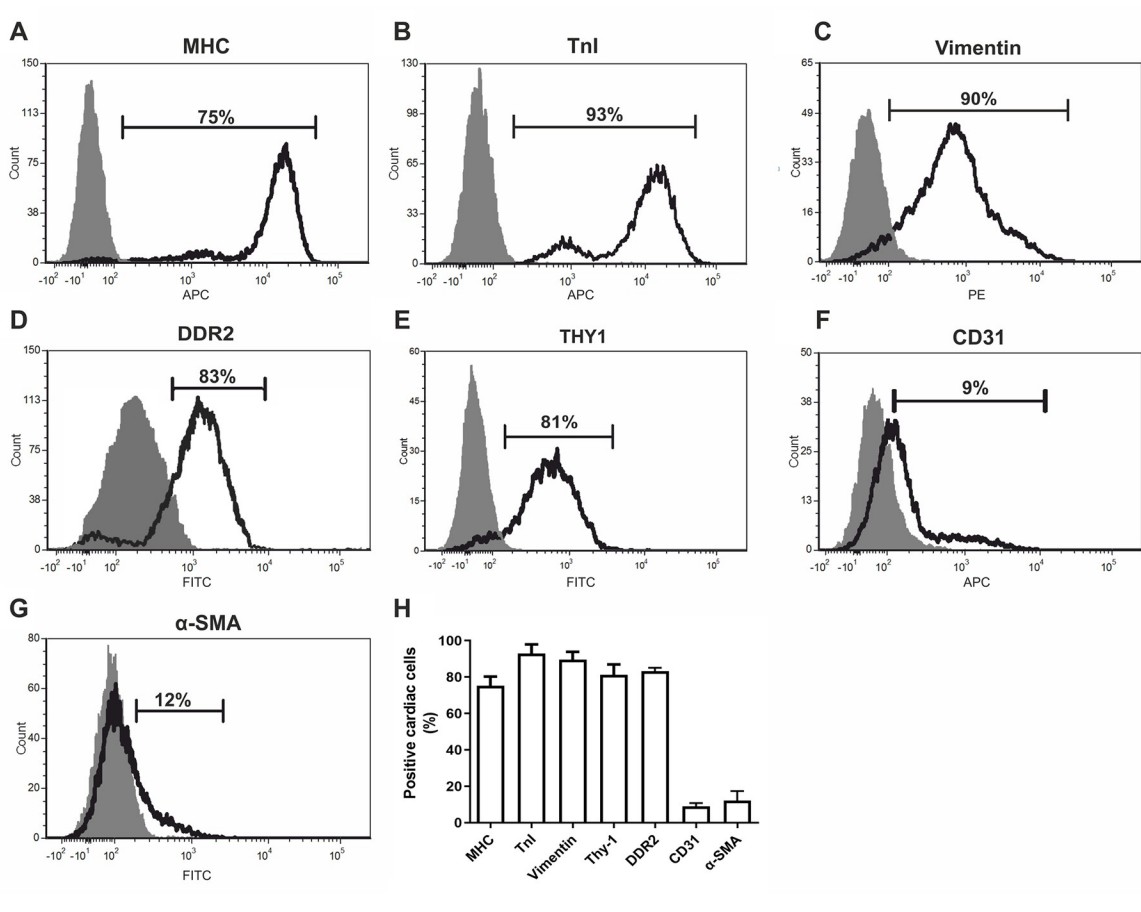

**Fig 3.**

the markers of cardiomyocytes, 75% of cells were MHC[+] (n = 21, SEM ±1.40) (Fig 3A) and 93% were cTnI[+] (n = 14, SEM ±1.5) (Fig 3B), suggesting a proportion of cTnI[+] cardiac cells are non-myocytes. For cardiac fibroblast markers, 90% of cells were vimentin[+] (n = 18, SEM ±1.2) (Fig 3C), 83% were DDR2[+] (n = 4, SEM ±0.6) (Fig 3D) and 81% were THY1[+] (n = 15 SEM ±1.40) (Fig 3E). For endothelial and smooth muscle cell markers, 9% of cells were CD31[+] (n = 12, SEM = ±0.54) (Fig 3F) and 12% were α-SMA[+] (n = 7, SEM = ± 1.9) (Fig 3G), respectively. These data suggest that cTnI is not cardiomyocyte specific and vimentin, DDR2 and THY1 are not fibroblast specific. A complete list of all single marker flow cytometry data is displayed in S3 Table.

Interpreting each flow cytometry dot plot for cTnI and vimentin revealed two distinct populations of positive cells based on their relative fluorescence intensity (high and low expressing) (Fig 4). On average, 11% of cells were VIM[High] (SEM ±1.01) and 78% were VIM[Low] (SEM ±0.54) (n = 7) (Fig 4A and 4B). On average, 79% of cells were cTnI[High] (SEM ±2.41) and 19% were cTnI[Low] (SEM ±2.1) (n = 14) (Fig 4C and 4D). The fluorescence intensity of the cTnI[Low] population is 1000 times greater than the negative and unstained (S1 Fig) controls, confirming this cell population does not represent a cTnI negative population. Furthermore, the fluorescence intensity of the cTnI[High] population is 10 times greater than the cTnI[Low] population, evidence that these represent two distinct populations. High and low expressing populations were also detected for cardiac troponin subunit T (cTnT (TNNT2)) (S4 Fig), with 28% of cells cTnT[Low] (SEM ±1.83) and 64% of cells cTnT[High] (SEM±0.3) n = 2. As a previously reported marker of cardiomyocytes, we predicted that cTnI[High] cells represented the cardiomyocyte population. To determine whether cTnI[Low] cells could represent a non-myocyte population, we performed flow cytometry analysis on cells of the fetal aorta/pulmonary artery, which are absent of cardiomyocytes. Cells from these tissues expressed cTnI at the same fluorescence intensity as the cTnI[Low] cardiac population, confirming cTnI is expressed at lower levels in non-myocyte cells types (Fig 4E).

## Dual marker flow cytometry and immunohistochemistry of fetal human heart reveal expression of vimentin at low levels in cardiomyocytes

Dual flow cytometry can determine the expression levels of two markers simultaneously within cell populations, enabling further interrogation of the cellular composition of the two cTnI[+]

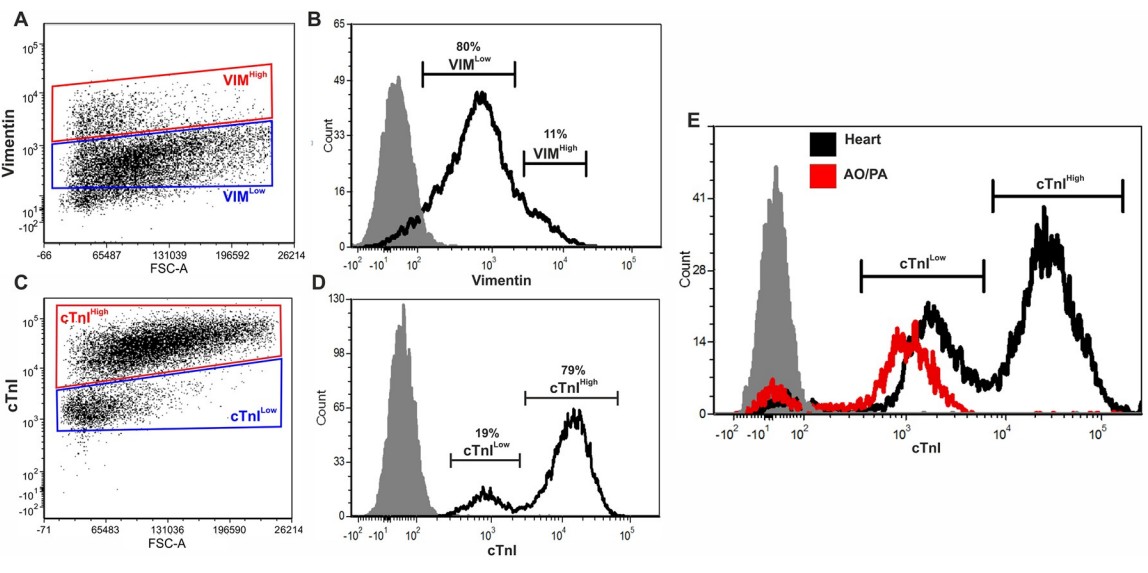

**Fig 4.**

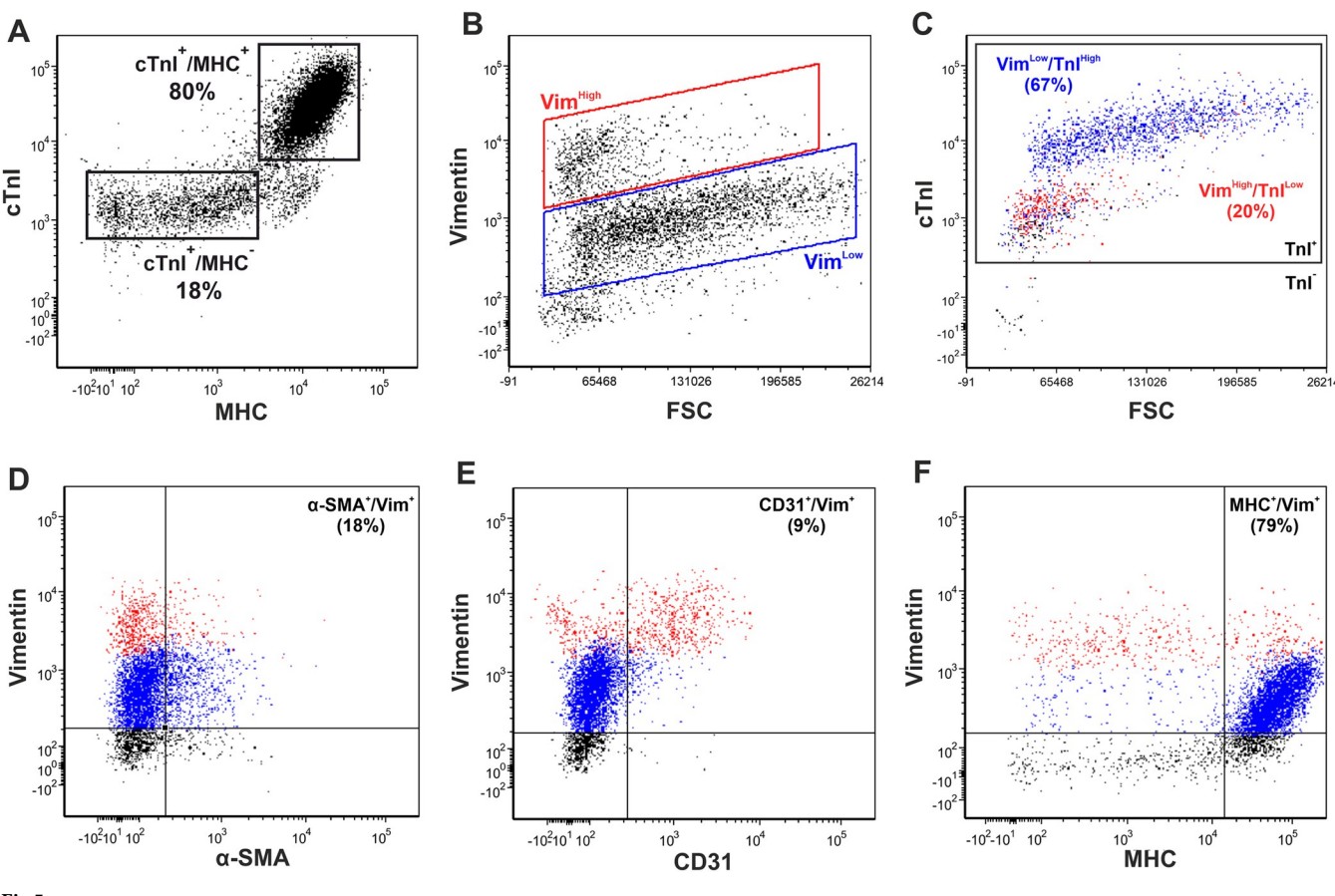

**Fig 5.**

populations. Cardiac cells were co-stained with cTnI and MHC and analysed by flow cytometry. cTnI$^{High}$ cells were also MHC$^+$ (80% of cells, SEM ±2.6) whilst cTnI$^{Low}$ cells were MHC$^-$ (18% of cells, (SEM ±2.64) (n = 5) (Fig 5A), supporting the theory that cTnI$^{High}$ cells are cardiomyocytes. To investigate the identity of vimentin positive populations, cardiac cells were co-stained with vimentin and cTnI, α-SMA, CD31 or MHC. VIM$^{High}$ and VIM$^{Low}$ cells were gated (Fig 5B) and plotted against these markers (Fig 5C–5F). The majority of VIM$^{Low}$ cells (blue) expressed cTnI at higher levels, whilst VIM$^{High}$ cells (red) expressed cTnI at lower levels (Fig 5C). On average, 67% of cardiac cells were VIM$^{Low}$/cTnI$^{High}$ (SEM ±3.38) and 20% were VIM$^{High}$/cTnI$^{Low}$ (SEM ±2.53) (n = 4). 18% of cells were α-SMA$^+$/VIM$^+$ (n = 2, SEM ±6.37), with the majority of this population expressing vimentin at lower levels (blue) (Fig 5D). 9% of cells were CD31$^+$/VIM$^+$ (n = 8, SEM ±0.84), with the majority of these cells expressing high levels of vimentin (red) (Fig 5E). 79% of cells were MHC$^+$/VIM$^+$ (n = 7, SEM ±1.88), with the majority of this population expressing low levels of vimentin (Fig 5F). Together, these data suggest that whilst fetal cardiac endothelial cells express vimentin at higher levels, fetal cardiomyocytes and cardiac smooth muscle cells express vimentin at lower levels. In support of these findings, immunohistochemistry analysis of fetal human heart tissue confirmed expression of vimentin at high levels (closed arrowhead) and lower levels (open arrowhead) in the myocardium (S5 Fig).

To further confirm the flow cytometry data, dual immunohistochemistry was carried out on fetal human heart tissue (Fig 6). Co-expression of cTnI and vimentin was seen in the

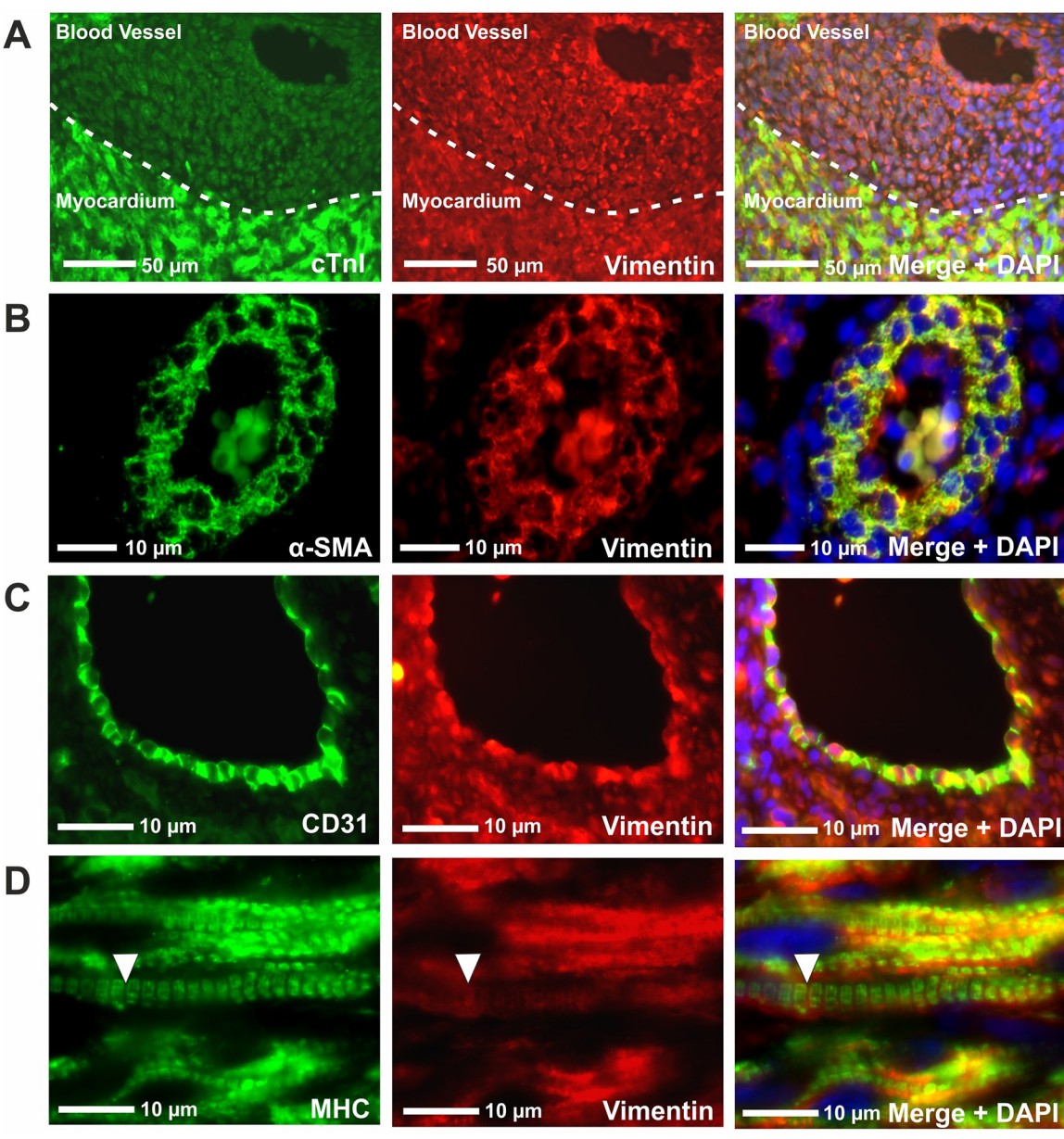

**Fig 6.**

myocardium and the cells of large blood vessels (the demarcation border between vessel and myocardium was defined by tissue structure and nuclei density and not cTnI expression) (Fig 6A). However, the fluorescence intensity of cTnI in the cells lining the blood vessel was much weaker relative to the myocardium, correlating with the flow cytometry data that showed high cTnI expression in myocytes and low cTnI expression in non-myocytes (Fig 6A). Colocalisation of α-SMA and vimentin was observed in coronary vessels (Fig 6B). Similarly, vimentin and CD31 co-stained the endothelial cells lining the cardiac blood vessels (Fig 6C). Co-localisation of vimentin with MHC in the myocardium supported the flow cytometry data that fetal human cardiomyocytes express vimentin (Fig 6D). Expression of DDR2 co-localised with MHC[+] cardiomyocytes and MHC[-] cells of the aorta/pulmonary artery vessels (S6A Fig).

CD31⁺ endothelial cells and α-SMA⁺ smooth muscle cells lining blood vessels also co-stained for DDR2 (S6B and S6C Fig). Dual staining of CD31 and α-SMA showed expression in distinct regions of the vessels, confirming specificity of these antibodies (S7 Fig). These findings indicate DDR2 is expressed in myocytes as well as and non-muscle cardiac cell types during early development.

## Troponins TNNI3 and TNNT2 are expressed in non-myocyte cells of the human fetal heart

Our flow cytometry data revealed expression of the cardiomyocyte marker cTnI in non-myocyte cell populations. To investigate this further, fibroblasts were isolated and cultured from fetal human cardiac ventricular tissue using explant migration, as previously described[9], which provided a non-cardiomyocyte cell population that could be used to assess the expression of key cardiac cell markers at the gene and protein level. To investigate the transcription of genes in myocyte and non-myocyte populations, RT-PCR was performed to detect marker transcripts, comparing whole fetal heart with isolated primary fetal ventricular fibroblast cells. Fetal heart tissue expressed *VIM (Vimentin)*, THY1, DDR2, *MYH6 (α-MHC)*, *TNNI3 (cTnI)*, *TNNT2 (cTnT)*, *PECAM-1 (CD31)* and *ACTA2 (α-SMA)* (Fig 7A). Ventricular fibroblasts were absent of *MYH6* and *PECAM-1* (Fig 7B) but expressed *TNNI3 and TNNT2*, further evidence for the expression of these genes in non-myocyte populations. The absence of *MYH6* transcripts demonstrated that cardiomyocytes had not migrated from the explants and thus

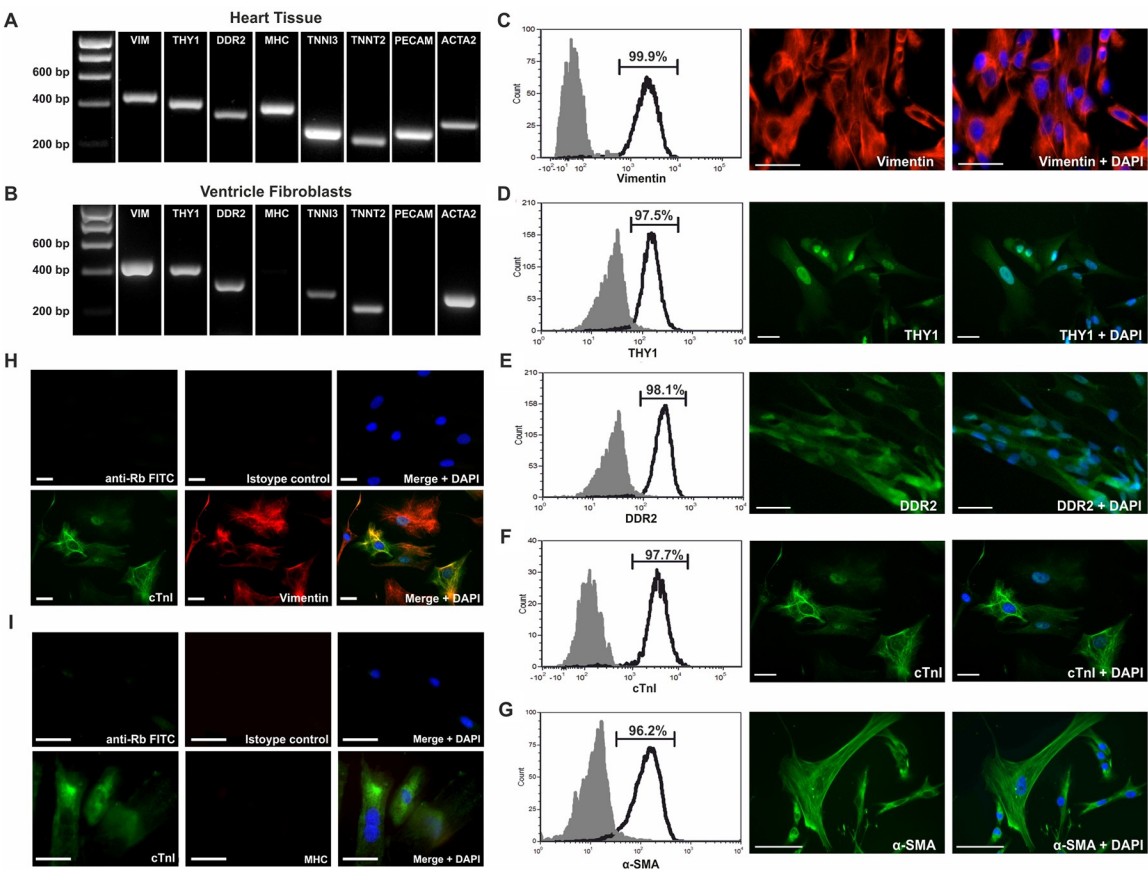

**Fig 7.**

excluded cardiomyocyte contamination as a cause of *TNNI3* and *TNNT2* transcript detection. Furthermore, the absence of CD31 transcripts also confirmed that the isolated fibroblasts were not contaminated by endothelial cells.

## Cardiac fibroblasts are enriched for Vimentin, THY1, DDR2, cTnI and α-SMA

To investigate the expression of key cardiac cell markers at the protein level in a non-myocyte population, flow cytometry and immunocytochemistry were performed on cultured fetal human ventricular fibroblasts. Fibroblasts were stained for Vimentin, THY1, DDR2, cTnI and α-SMA. Flow cytometry revealed enrichment of all proteins in these cells (>96% positive) (Fig 7C–7G). Dual immunocytochemistry of cultured ventricular fibroblasts showed co-expression of vimentin and cTnI. (Fig 7H). The absence of MHC expression in cTnI positive fibroblasts further confirmed this finding is not as a result of cardiomyocyte contamination (Fig 7I). Interestingly, the staining pattern of cTnI in the ventricular fibroblasts is distinct from that observed in mature cardiomyocytes, with a lack of sarcomeric organisation. These data provide further evidence that cTnI is expressed in non-myocyte cells of the fetal heart.

## Discussion

This study aimed to define the cellular composition of the early fetal human heart, an objective that is impaired by the lack of specific markers for some key cardiac cell types. As such, this study also evaluated the expression and specificity of a range of commonly used cardiac markers to help define novel marker profiles for cardiac cell populations during early development. Immunohistochemical analyses of cardiomyocyte markers MHC and cTnI showed defined sarcomeric staining in the myocardium of fetal heart tissue confirming their expression in cardiomyocytes. Notably, flow cytometry analyses showed a greater percentage of cardiac cells expressing cTnI (93%) compared to MHC (79%), suggesting cells other than cardiomyocytes may express cTnI. Flow cytometry and immunocytochemistry analyses were able to confirm the expression of cTnI at a lower level in non-myocytes (MHC$^-$) and higher level in cardiomyocytes (MHC$^+$). Furthermore, TNNT and TNNI gene expression in ventricular fibroblasts was established by RT-PCR, further supporting the presence of troponins in non-myocyte cells. Our results are strengthened by the recent findings from Cui *et al.*, [31] identifying positive immunostaining of cTnT in cardiac fibroblasts from 17 gestational week fetal cardiac tissue. Furthermore, transcriptional analysis of fetal cardiac populations at the single cell level has revealed expression of TNNT3 in a subset of endothelial cells, further evidence that troponins are not specific to muscle cells [32]. The non-striated pattern of expression of cTnI in cardiac fibroblasts observed in this study suggest a non-sarcomeric organisation and potential non-sarcomeric role, however, the exact function of cTnI in these non-contracting cells remains unclear. Other previous studies have shown the expression of specific isoforms of troponin, as well as other contractile proteins, including myosin and tropomyosin, in smooth muscle cells [33–35], suggesting that cardiac troponin may also have non-cardiomyocyte-specific isoforms in the context of cardiac development. Indeed, the role of TnT in multiple cellular processes beyond contraction in skeletal muscle [36], including transcription-related processes, and the recognised role of troponins in cancer [37], has established that troponins' effects are multifaceted and isoform-dependent. Overall, these data suggest troponin proteins are not specific markers of cardiomyocytes in the developing human heart, which should be considered when evaluating the use of these proteins to identify and isolate cardiomyocytes and may further be valuable in the purification and differentiation of PSC-CMs.

The mesenchymal marker vimentin is a commonly used fibroblast marker; however, we showed approximately 90% of fetal cardiac cells are positive for vimentin, suggesting cells other than fibroblasts express this protein. Flow cytometry analysis showed expression of vimentin at low levels in MHC$^+$ cardiomyocytes (MHC$^+$/VIM$^{Low}$) and high levels in MHC$^-$ non-myocyte cells. We were able to confirm that cardiac fibroblast cells isolated from ventricular tissue are enriched for vimentin relative to total heart (ventricle/atria). We also confirmed that endothelial cells express high levels of vimentin, supporting what has previously been described in the adult heart [22]. These findings are consistent with a previous study that identified weak expression of vimentin in some cardiomyocytes of 9–14 pcw human hearts and demonstrated that with increasing fetal age vimentin expression in cardiomyocytes decreases and desmin expression increases [38]. It remains unclear the precise role of vimentin in fetal cardiomyocytes, as well as the reason for its absence in adult cardiomyocytes. The collagen receptor DDR2 has been identified as a more specific marker of adult cardiac fibroblasts [39], however, our results showed its expression in cardiomyocytes, endothelial cells and smooth muscle cells of the developing heart.

Flow cytometry of fetal cardiac cells stained for the cardiomyocyte marker MHC revealed a large population of highly expressing cells (fluorescence intensity $>10^4$) alongside a very small, broader peak of cells expressing MHC at lower levels. Whilst both populations are gated as MHC$^+$, we chose not to interpret this small peak as a distinct population as it did not exhibit the same characteristics as the cTnI or vimentin low populations. However, we cannot exclude the possibility that this peak may represent a distinct population of positive cells expressing MHC at lower levels.

Utilising multiple experimental approaches and cellular markers, our results offer an estimate of the relative cellular composition of the human fetal heart and distinct marker profiles to distinguish between fetal cellular populations. Our analysis supports the use of three markers to define the fetal human cardiomyocyte population: 'MHC$^+$/cTnI$^{High}$/VIM$^{Low}$ ' represents the fetal human cardiomyocyte population, which we estimate to comprise 75–80% of total cardiac cells. This is consistent with other reports, which have shown cardiomyocytes to be the largest cell population in the fetal heart [31, 32]. In contrast, current estimates of the adult human heart suggest non-myocytes to be the dominant population [4, 6, 7]. Specifically, using nuclei analysis of human tissue, Pinto *et al* showed that cardiomyocytes occupy 31% of the adult cardiac population, whilst Tucker *et al* conducted single nuclear RNA sequencing to reveal an adult cardiomyocyte population comprising 36% of cells [40]. In the fetal heart, Banerjee *et al.*, [16] showed cardiomyocytes to occupy around 65% of the late-stage embryonic mouse heart, whilst in human fetal tissue, Suryawanshi *et al.*, [32] used single cell sequencing to reveal cardiomyocytes account for nearly half of the total cell composition. Furthermore, Cui *et al.*, used single cell transcriptomics to show cardiomyocytes make up the largest population of cells in the embryonic heart [31]. Mitosis of differentiated cardiomyocytes in the developing heart is well documented and is responsible for cardiac morphogenesis and organogenesis in utero [41]. From two weeks after birth, cardiomyocyte proliferation is significantly reduced in the mammalian heart as these cells enter cell cycle arrest, with continued cardiac growth reliant on hypertrophy of pre-existing cardiomyocytes [42, 43]. Therefore, cardiomyocytes occupying the largest cellular population during cardiac development likely reflect the proliferative nature of fetal cardiomyocytes relative to adult cardiomyocytes.

In this study, the non-cardiomyocyte population was estimated to comprise 20–25% of total cardiac cells and exhibit an MHC$^-$/cTnI$^{Low}$/VIM$^{High}$ marker profile, of which approximately 10% are endothelial cells (CD31$^+$/VIM$^{High}$). Due to the lack of a fibroblast-specific marker, it is difficult to accurately determine the percentage of fibroblasts and smooth muscle cells, although based on the percentage of endothelial cells within this non-myocyte

population, it is likely in the region of 10–15%. Indeed, we found 11% of cells were $VIM^+/\alpha\text{-}SMA^+$ which may represent a cell population with a fibroblast and smooth muscle cell phenotype. Smooth muscle cells have been shown to transcriptionally cluster with fibroblasts and exhibit fibroblast features, including expression of extracellular matrix genes, suggesting a common developmental lineage [31, 32], which would impede discrimination between these cell types if the phenotypes have yet to become distinct [31, 44]. However, the nature of *in vitro* culture conditions can initiate primary fibroblasts to take on a smooth muscle phenotype and therefore we cannot conclude that cardiac fibroblasts *in vivo* exactly mimic the expression patterns we observed.

This study focused on the four main cell types of the heart and therefore we cannot comment on the contribution of other important cardiac cells types to the cellular composition of the fetal human heart, including mural and immune cells. Further investigations are required to determine the exact composition of the non-myocyte population during early cardiac development. Due to the lack of reliable surface markers for fibroblasts that would enable cell sorting for downstream RNA and protein applications, the isolation and culture of primary fibroblasts is currently the only viable option to investigate this cell population.

The results from our study suggest the marker profiles of fetal cardiac cells are distinct from those of adult cardiac cells. The phenotypic similarities between the mid-gestation human fetal heart and pluripotent stem cell-derived cardiomyocytes suggests our data could be useful when purifying and characterising these cells. Furthermore, the marker profiles identified could potentially be used for future studies to determine how the ratio of cardiomyocytes to non-cardiomyocytes changes throughout fetal development.

## Supporting information

**S1 Raw images.**
(PDF)

**S1 Fig.** (A) Representative flow cytometry side scatter and forward scatter dot plot of fetal human cardiac cells showing gating of the cellular population for downstream analyses. (B) Representative flow cytometry histogram of unstained heart cells at the APC fluorophore wavelength (max excitation 650nm, max emission 661nm). (C) Representative flow cytometry histogram of unstained heart cells at the PE fluorophore wavelength (max excitation 566nm, max emission 574nm). (D) Representative flow cytometry histogram of unstained heart cells at the FITC fluorophore wavelength (max excitation 490nm, max emission 525nm).
(TIF)

**S2 Fig. Representative flow cytometry gating strategy used to determine positive populations of cardiac cells for each marker.** Dot plots with fluorescence on the Y axis and FSC on the X axis were used to determine positive populations. The distribution of the negative control population (isotype or secondary antibody only) was used to draw the positive gates. Cells expressing fluorescence levels above the negative controls were marked as positive.
(TIF)

**S3 Fig. Representative immunohistochemistry images of antibody negative controls in fetal human heart tissue.**
(TIF)

**S4 Fig. Representative flow cytometry histogram of fetal human heart cells stained for cTnT showing two distinct populations of cTnT$^{High}$ (64%) and TnT$^{Low}$ (28%) expressing**

**cells (n = 2).** Grey histogram peaks represent antibody controls.
(TIF)

**S5 Fig. Immunohistochemistry of fetal human heart tissue showing expression of Vimentin at high levels (closed arrowhead) and lower levels (open arrowhead) in the myocardium.** DAPI was used as a counter stain for cell nuclei.
(TIF)

**S6 Fig.** DDR2 dual immunohistochemistry of fetal human heart tissue: (A) DDR2 and MHC expression at the boundary (dotted line) of a blood vessel wall and the myocardium. (B) DDR2 and α-SMA expression in a cardiac blood vessel (C) DDR2 and CD31 expression in a cardiac blood vessel. Myo = myocardium. AO/PA = aorta/pulmonary artery. DAPI was used as a counter stain for cell nuclei.
(TIF)

**S7 Fig. Dual immunohistochemistry of fetal human heart tissue expressing α-SMA and CD31 showing expression in distinct regions of the blood vessel.** α-SMA expression is seen in the smooth muscle vessel lining and CD31 expression is seen in the endothelial inner lining of the vessel. DAPI was used as a counter stain for cell nuclei.
(TIF)

**S1 Table. List of primary and secondary antibodies, their working concentrations and their corresponding controls.**
(DOCX)

**S2 Table. Primer sequences and amplicon sizes for detection of cardiac cell marker mRNA transcripts using RT-PCR.**
(DOCX)

**S3 Table. Complete list of flow cytometry data showing tissue age, percentage of positive cells recorded for each marker and number of samples analysed.**
(DOCX)

## Author Contributions

**Conceptualization:** Jennifer M. Dewing, Vinay Saunders, Ita O'Kelly, David I. Wilson.

**Data curation:** Jennifer M. Dewing.

**Formal analysis:** Jennifer M. Dewing.

**Funding acquisition:** David I. Wilson.

**Investigation:** Jennifer M. Dewing.

**Methodology:** Jennifer M. Dewing, Vinay Saunders.

**Project administration:** David I. Wilson.

**Supervision:** Ita O'Kelly, David I. Wilson.

**Validation:** Jennifer M. Dewing.

**Visualization:** Jennifer M. Dewing.

**Writing – original draft:** Jennifer M. Dewing, David I. Wilson.

**Writing – review & editing:** Jennifer M. Dewing, Vinay Saunders, Ita O'Kelly, David I. Wilson.

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
