## [Decision Letter · Decision Letter 0]

23 Nov 2021

PONE-D-21-32716Defining Cardiac Cell Populations and Relative Cellular Composition of the Early Fetal Human HeartPLOS ONE

Dear Dr. Dewing,

Thank you for submitting your manuscript to PLOS ONE. After careful consideration, we feel that it has merit but does not fully meet PLOS ONE’s publication criteria as it currently stands. Therefore, we invite you to submit a revised version of the manuscript that addresses the points raised during the review process.

We look forward to receiving your revised manuscript.

Kind regards,

Federica Limana

Academic Editor

PLOS ONE

Journal Requirements:

[This work was funded by the MRC DTA Capacity Building Studentship G1000406. The human embryonic and fetal material was provided by the Joint MRC/Wellcome Trust grant #099175/Z/12/Z Human Developmental Biology Resource (HDBR)]

 [J.M.D received DTA Capacity Building Studentship funding from the Medical Research council UK: thttps://mrc.ukri.org/skills-careers/studentships/how-we-fund-studentships/doctoral-training-partnerships-dtps/

Grant number: G1000406

The funders had no role in study design, data collection and analysis, decision to publish, or preparation of the manuscript

VS, IOK and DIW were not directly funded by this award.]

[The authors have declared that no competing interests exist].  

We note that one or more of the authors are employed by a commercial company: Immunocore Ltd, Abingdon, Oxford, United Kingdom. 

Reviewers' comments:

Reviewer's Responses to Questions

**Comments to the Author**

1. Is the manuscript technically sound, and do the data support the conclusions?

Reviewer #1: No

Reviewer #2: Partly

Reviewer #3: Partly

2. Has the statistical analysis been performed appropriately and rigorously? 

Reviewer #1: N/A

Reviewer #2: I Don't Know

Reviewer #3: N/A

3. Have the authors made all data underlying the findings in their manuscript fully available?

Reviewer #1: No

Reviewer #2: Yes

Reviewer #3: No

4. Is the manuscript presented in an intelligible fashion and written in standard English?

Reviewer #1: Yes

Reviewer #2: Yes

Reviewer #3: No

5. Review Comments to the Author

Reviewer #1: Fig1

Multiple issues.

Issue 1

Not enough information is provided to follow what the authors did. Under the section ”antibodies for flow and IHC”, a mix of directly conjugated and unconjugated primary antibodies are given. It is unclear exactly which antibodies were used for IHC and whether the same antibodies were used for both IHC and flow. Only one antibody for each antigen is listed, suggesting the same was used for flow and IHC yet the methods state that secondary antibodies were used for IHC staining, suggesting unconjugated primaries were used. This is unclear.

We need a table to show exactly which primaries/secondaries were used for each experiment (IHC and flow), with catalogue supplier numbers, dye conjugations, paired control isotype antibodies etc.

Issue 2

As far as I can determine no controls were performed to test the specificity of the antibodies used. Both positive and negative controls are needed to support the conclusions.

We need to see negative controls omitting primary antibody to show that the fluorescence reported does not result from autofluorescence.

The evidence for specificity given in the paper comes simply from IHC data showing expression patterns resembling that expected. In the case of the cardiomyocyte markers TnI and MHC, striations can be seen within the cells, giving confidence that the stained cells are indeed striated muscle. Endothelial and smooth muscle cell markers appear to be staining blood vessels as expected (although double labelling is needed to show that the expression does not overlap). However, it is more difficult to conclude that the fibroblast markers are staining fibroblasts as no clear anatomical features have been identified.

We need to see a positive control for these antibodies. Either a Western blot or dot blot to demonstrate binding to the correct protein is needed. Alternatively, staining of a known cell line would provide confidence in cell identification.

Fig 2

Multiple issues with this key figure.

Issue 1

The interpretation of the data (presence or absence of marker) is very much dependent on how the raw data is gated. Although the methods state that the position of the gate was set based on the control data this does not seem to always be the case. In fact, positioning varies between different panel in Fig2. In some cases the gate is positioned close to the negative control peak whereas in others it is some distance away, excluding some datapoints that would otherwise be classed as positive.

Issue 2

Not enough information is provided to understand what was done, and as a result we cannot conclude that the data is reliable.

As stated above, it is unclear whether the antibodies used are same as those used for IHC.

The antibody staining procedure is not given in the methods. The study employs a mix of directly dye-conjugated and unconjugated antibodies, which require different methods (the latter require a fluorescent secondary) but this is not clearly explained. E.g. was unbound antibody washed off? Were primary and secondary applied sequentially?

A detailed staining protocol needs to be provided.

Issue 3

The controls used are not the same for all antibodies.

The methods are unclear, but for directly conjugated antibodies the authors appear to have used an isotype control, while for unconjugated primary antibodies they use a secondary antibody. Are these equivalent? The level of fluorescence of different controls appears to vary between graphs by a factor of 10.

The legend to Fig2 states that the grey peak indicates “isotype/negative controls”. What does this mean? What is a negative control?

We need to know exactly what was used as the negative control for each antibody, and this should be included in the above table of antibodies.

Issue 4

The quality of the data varies considerably between different antigens. The graphs for DDR2 (Fig2e) and THY1 (Fig2f) show high quality data. In these cases, two peaks are seen. There is a peak which overlaps with the negative control, indicating cells that are negative for the marker, and then a second peak clearly distinct from and not overlapping with the first, indicating cells positive for the marker. Data for other markers is more questionable, as detailed below.

Issue 5

There are no positive controls.

Do the antibodies work and bind to the expected protein? Again, positive controls are needed (Western, dot blot, or known cells) to prove specific binding.

2a. The FSC x SSC plot used to identify cells needs to be shown.

2b. The gate has not been applied as stated in the methods and is instead shifted a long way from the negative control such that some fluorescent cells ( in a peak at about 10-3) are excluded. The authors appear to assume that this lower-intensity peak is a negative signal, even though it is clearly distinct from the negative control. It appears to me that there is a problem with the negative control in this and other graphs, but detailed information on the control used is missing from the methods and figure legend.

2c. This graph is very similar to that shown in Fig 2b, but oddly the gate has been placed in a different position. In this case, the gate includes both high and low intensity peaks. One of these two panels must be wrong. Data shown in Fig 3e (discussed below) suggests to me that the lower intensity peak represents cells negative for this marker, but this is not how the authors interpreted this data.

2d. There is a large low-intensity peak and a smaller high-intensity peak, partly overlapping. The gate includes both peaks. Again, I think the lower-intensity peak represents negative cells.

2e. No issues

2f. No issues

2g. The graph shows a high low-intensity peak, partly overlapping with the negative control and a smaller high-intensity peak, which is distinct. The gate has been placed at the edge of the negative control peak, and therefore includes all of the high-intensity cells and some of the low-intensity. I think this is wrong. The lower intensity peak most likely represents the cells negative for the marker and the difference between this peak and the control is likely due to a problem with the unspecified control.

2h. Only a single peak is seen, which is shifted relative to the control. There may be a very low higher-intensity peak to the right, but this overlaps with the first. Issues with gate positioning in 2g apply here as well.

2i. Data is derived from above, and therefore needs to be revised.

Fig 3

3a, 3c. It is unclear to me why the boxes shown on these graphs are not rectangles with 90° angles. They appear to be based on the fluorescence gates shown in Fig 3b and 3d, but these gates are at a fixed fluorescence level, not one that changes with cell size (FSC).

3b,3d. The two peaks shown on each graph are labelled as low and high level expression, but these could be interpreted as positive and negative expression if the control is wrong (discussed above).

3e. The data shows that cells from vascular tissue (which does not contain striated muscle) showed the same level of fluorescence as the lower TnI peak in heart cells. Troponin is used only by striated and not by smooth muscle. The authors conclude from these data that a population of cells expressing low levels of TnI exists in both heart (striated muscle) and vascular tissue (smooth muscle). An alternative interpretation of these data, and to my mind the more likely explanation, is that this peak represent cells negative for TnI, even though it is shifted relative to the negative control. To answer this we need more information on what the negative control is, and preferably also data for a negative control cell type.

Fig 4

Issue 1

In order to identify double or triple labelled cells using flow sorting, it is necessary to use fluorescent dyes whose emissions spectra do not overlap, otherwise signal will bleed through between channels. The authors do not tell us which dyes were used to perform these experiments and therefore it is impossible to assess whether the data is accurate.

Issue 2

Furthermore, the accuracy of these data depends on the correct placing of the gates shown in Fig 2. As stated above, I believe that some of the gates used were incorrect.

4a. The data shown in Fig 3e suggests that the lower TnI peak represents cells negative for TnI. If so, then the population here labelled as Tni + / MHC – are actually negative for both markers suggesting they are simply the non-cardiomyocyte population of the heart.

4b. Again, I think the boxes should be rectangles if based on fluorescence gates.

4c. The same issues that apply to 4a apply here. If the cells labelled as low expression are in reality negative for the marker then the result is more logical: we have one population of troponin positive cardiomyocyes and one population of vimentin positive fibroblasts.

4d-4f. Why is this data presented in a different manner to 4c? Surely all graphs should be either marker x FSC or marker1 x marker2. Secondly, the colour-coding of these datapoints is unnecessary – it is supposed to show populations expressing high and low Vim, but we can judge for ourselves whether such populations exist by looking at the y axis. Again the gating here is dependent on Fig 2 being correct.

Fig 5

Issue 1

Not enough information is provided to follow what the authors did. The legend does not tell us what tissue is shown nor how the images relate to each other.

Issue 2

All of the issues mentioned in Fig1 also apply here and make these data unreliable.

The co-localisation of vimentin with other markers reported for both IHC and flow data could be explained by non-specific binding of the vimentin antibody. Controls are needed to disprove this null hypothesis.

Fig 6

Fig 6 is offered as supporting evidence for the hypothesis that a population of troponin-expressing fibroblasts exists in the heart. To show this, the authors dissected out pieces of heart tissue and then selected for cells which migrated out of the heart, which they describe as fibroblasts. They then used a crude endpoint RT-PCR to demonstrate co-expression of TnI/TnT with Vim in these cells.

There are a few issues here.

Issue 1

The cells isolated have not been characterised and have not been shown to be a pure population of fibroblasts.

We need IHC data to show that this is a homogenous population in which all cells either express or do not express a given marker (is this what is shown in Fig 7? – unclear).

Issue 2

Under certain conditions, cardiac fibroblasts can trans-differentiate into cardiomyocytes. It is possible that migratory fibroblasts have begun to trans-differentiate into another cell type. What is the evidence that these are the same cell type as exist in the foetal heart?

Issue 3

The RT-PCR data is not quantitative, a single contaminating cell could give a false positive.

Fig 7

It is unclear whether these are the same cells used in Fig 6. Again, more information is needed. What passage number are these and those in Fig 6? Are they from the same heart?

IHC – we need a DAPI co-stain in order to assess whether there are cells negative for each marker. Double labelling with vimentin would also be useful to support the hypothesis that fibroblasts express troponin.

Flow- the level of fluorescence of the positive peak appears to be different to that shown in Fig 2 for some panels. If this is the same antibody how can that be the case? We need to see flow data for MHC and CD31 to confirm the negative RTPCR results and therefore give more confidence in the positive data shown.

Minor points

1. The authors should give the official names of the proteins studied for clarity (as listed by NCBI). In some cases multiple isoforms of a protein exist, and the names used by the authors do not make clear the specific isoform referred. For example, there are three Troponin I proteins and the one normally expressed in heart is TNNI3. Similarly, there are multiple myosin heavy chains and the one referred to is MYH6.

2. Line 72 “Thy1… as a surface protein it has the advantage over vimentin and DDR2 in that it does not require cell permeabilisation and can therefore be used for fluorescent activated cell sorting”. DDR2 is also a transmembrane protein that can be used in FACS. Furthermore, the authors go on to show FACS data for many intracellular proteins.

3. typo line 93 - estimare

Reviewer #2: The authors explored the cellular composition of the fetal heart in a brief descriptive study that flows well. The aim of the study is to evaluate the specificity of certain cell markers in identifying the cell composition of the fetal heart. The authors explain the challenges of identifying cell-specific markers, especially for fibroblasts, in the adult myocardium, but did not provide robust suggestions of cell-specific markers in the fetal heart. Here are my recommendations that may help improve the current manuscript.

1-Please provide the detailed methodology for the isolation of cardiac fibroblasts and cardiomyocytes.

2-Please provide more details on the antibodies that you used in your immunofluorescence and FACS, including antibody dilutions, incubation time, etc.

3-The authors show that "the fibroblast markers showed extensive staining throughout the heart tissue" One would question the specificity of the antibody used and the technique of your IHC staining. Please include any possible positive or negative controls (including images from isotype-matched control staining for example).

It would also be more reliable if you dual stain each of the fibroblast markers with a GFP staining in PDGFRa-GFP mice (commercially available).

4-The quality of the immunofluorescence images in figure 1 should be enhanced. More examples should be provided from different regions of the myocardium. Do you see differential labeling of specific markers in different regions of the LV (sub-endocardium, sub-epicardium, or mid-myocardium)? Did you observe differential labeling in RV or the atria compared to LV?

5-Again, in the flow cytometry, one would need to see appropriate validation of the primary antibodies, and the appropriate negative controls (including the isotype control), and positive controls (for example, cells from fibroblast reporter mice).

6-With the challenges that you explained in identifying a fibroblast marker, how about mEF-SK4 as a marker of fibroblasts (Pinto, A. R. et al. Cir Res 2016).

7-"TnI-Low cells were MHC-", If these cells are not MHC+, are they fibroblasts, endothelial cells, or other cell types?

8-Staining of cultured fibroblast for α-SMA does not reflect the biology in the fetal heart. Cultured cardiac fibroblasts are under high tension thus they transform to a-SMA expression myofibroblasts.

9-Please tone down your conclusions and use non-speculative language as possible in your discussion, for example, "suggesting a role of troponins in non-muscle cells distinct from that of cardiomyocytes and skeletal muscle cells", "vimentin expression may be a property of cardioymyocytes of the developing heart only.".

10-Based on your current data, I wouldn't speculate that cardiomyocytes are "estimated to comprise 75-80% of total cardiac cells.". Additionally, what about other cell types other than endothelial cells, and fibroblast that are not included in your percentages of fetal cardiac cell populations (including pericytes, macrophages, etc)?

11-Please correct the minor typos as in line 93 "estimare", line 189, etc.

Reviewer #3: The manuscript “Defining Cardiac Cell Populations and Relative Cellular Composition of the Early Fetal Human Heart” tries to analyze commonly used markers for the main cardiac cell types, such as cardiomyocytes, cardiac fibroblasts, endothelial cells and smooth muscle cells, in human fetal hearts (< 12 pcw ). Their findings are mainly based on antibodies used for Immunohistochemistry as well as flow cytometry analyses. However, they never tested specificity of these antibodies.

Major Comments

This directly leads to my main concern about this manuscript: Results and numbers from flow cytometry analyses seem plausible, however, there is no proof of specificity of the antibodies used. In addition, no complete gating strategy is shown. The authors should show (for example in a Supplemental figure) how gates were set in each dot plot of flow cytometry. If possible a positive control should be used for each antibody.

Further, it would be helpful to directly compare the findings to adult specimen. In the discussion, the authors talk about the differences in cell composition and marker expression between fetal hearts and adult hearts. But given the different antibodies used for detection of cell populations and different methods used for dissociation of tissues, such numbers might be highly variable. Please perform immunohistochemical stainings and flow cytometry analyses with the same antibodies on adult specimen.

The manuscript is unfortunately written in a rather long-winded, ponderous style, e.g. at p 3, the sentences beginning in line 49 or line 58ff. Please shorten and clarify sentences where possible.

References: The newest reference cited by the authors is from the year 2019. However, science is rapidly progressing, especially on the topic of cardiac development and fetal hearts since the heart is one of the most important organs and the one developing first in embryogenesis. References should be updated and newest publications, especially concerning single cell analysis of fetal human hearts should be included.

Specific Concerns

p2: Abstract

p2, line 26: …which remains largely unknown: here you should include something like “at least in humans”

p2, line 30: …Myosin Heavy Chain: you should specify if you mean the alpha or beta isoform

p2, line 30/31: you should use a consistent nomenclature to describe gene or protein names; (e.g. sometimes you only use the abbreviation, sometimes the complete name is written…).

p3: Introduction

p3, line 63: you should shortly discuss the different embryonic origins of fibroblasts which contribute to fibroblast heterogeneity

p4, line 72: DDR2 is also a surface Protein and is mainly found on the plasma membrane (by the way: this sentence should also be shortened, maybe build 2 sentences.)

p4, line 84: CD31 is not specific for endothelial cells (is also expressed in macrophages), alphaSMA is not specific for Smooth muscle cells but also expressed in activated fibroblasts. Please relativize that.

p4, line 93: estimate not “estimare”

p5: Materials and Methods section

the whole section should include more experimental details so that one is able to follow the experiments

a table with the specimen used should be included

p5, line 106: 70µm strainer: this is a size discrimination, did you check if you lose cardiomyocytes in this step? Or are fetal cardiomyocytes smaller than 70µm?

p5, line 109: heading should be “Fibroblast Isolation and Cell Culture”

p5, line 115: you should please note centrifugation speed in [g] not in rpm, since rpm is dependent on the rotor radius

p5, line 117: 35µm strainer: this is a size discrimination (even more than mentioned above line 106), did you check if you lose cardiomyocytes in this step? Or are fetal cardiomyocytes smaller than 35µm?

p5, line 133: as already noted above: how did you check specificity of antibodies? E.g. positive controls? Double stainings?

p5, line 135: as already noted in the abstract: is it alpha or beta MHC or does the antibody detect both isoforms?

p7, line 143: please note in that paragraph that results of the RT-PCR were visualized on an XXX% agarose gel and the fluorescent dye you used….

p7: Results

p7, line 157: TnT is not shown in Fig1A? Missing? Not stained?

p7, line 159: now you write that you stained alphaMHC. There are two isoforms of MHC: alpha and beta. The beta isoform is more relevant in fetal cardiac tissue while the alpha isoform is more typical for adult cardiac tissue. Could you please discriminate between the two isoforms? Are there antibodies that are specific for each isoform? Would be interesting to analyze the amount of beta versus alpha…

p7, line 161: since Thy1 is a surface protein, it should not be localized in the cytoplasm…. Vimentin should mark fibres of the cytoskeleton (However, I cannot recognize these patterns since the picture quality is so poor). In addition, there is DDR2 in the figure (not mentioned in the text).

p7, line 162: how do you know that these antibodies stain endothelial cells or smooth muscle cells? This is not the thing you can read from this picture. You only see that the antibodies stain cells from vessel walls.

p8, line 181/182 (legend for Fig 2I): Percentages of MHC and TnI are not the same as mentioned in the text???? Please correct.

p8, line 187/188: Please indicate percentages for TnT high and low expressing populations. By the way, the same holds true for MHC. Can you also add this into the Suppl Figure?

p8, line 188ff: You should do stainings for fetal aorta/pulmonary artery not only with TnI antibodies but also with Vimentin, TnT and MHC.

p9, line 213 (corresponding to Fig 4 C): Could you please include gates into this dot plot similar to the other dot plots in the same Figure? For Fig 4D: in the dotplot itself there should be aSMA/Vim not TnI/Vim, right?

p9, line 221: I know that you regard Vimention as a fibroblast marker. However, this marks cells from mesenchymal origin. Thus for a proof of principle you should also double stain Vimentin and DDR2 and Vimentin and Thy1.

p10, line 234: However, Vimentin intensity is not different in Fig5A (in the FACS you showed Vimlow for TnI high cells.

p11, line 255: Fetal heart tissue also expresses THY1 and DDR2; the gene name for aSMA is ACTA2 (in Fig 6 you should not use the protein names but the gene names which are sometimes different). Human gene names are written in capitals (e.g. THY1 not Thy1)

In Fig 6 you should show a negative control and an amplification control such as GAPDH or BetaActin for all the lines so that one can decide if the differences in the band intensities are really reflecting differences in gene expression and not only differences in amplification efficiency, etc.

p12, line 267: you should write “….in fetal cultured ventricular fibroblasts”

p12, line 268: THY1 and DDR2 are missing

p12, line 272: you should also stain for CD31 and MHC and check if cultured fibroblasts are negative for those antibodies (important for specificity of those antibodies).

Figures

The overall quality of pictures in the figures is not very good, especially the immunohistochemical stainings are rather blurry and overexposed. Please improve the quality of these images (Fig 1, Fig 5, Fig 7, Suppl Fig 2), e.g. by taking pictures with a confocal microscope.

The Figure Legends should also be improved. In the legends you should shortly describe what you see in the pictures, so that you can understand without reading the text.

p12 Discussion

p13, line 285: you cannot say that cTnI lower levels were confirmed in non-myocytes by RT-PCR, since you have no housekeeping gene in your RT PCR as a normalization control.

p14, line 315: why should there be myofibroblasts in a fetal heart? This is a cell type that is usual representing some kind of pathology in a heart…

In the discussion you should compare your results from human fetal hearts to results from murine fetal hearts and of course to state-of-the-art single cell studies on the cellular composition and gene expression of human fetal hearts.

6. PLOS authors have the option to publish the peer review history of their article (what does this mean?). If published, this will include your full peer review and any attached files.

Reviewer #1: **Yes: **Iain Dykes

Reviewer #2: **Yes: **Anis Hanna

Reviewer #3: No

---

## [Author Response · Author response to Decision Letter 0]

13 May 2022

We would like to thank the reviewers for their invaluable comments on this manuscript. We have now addressed all the points raised by the reviewers, which can be found in the submitted point-by point rebuttal and the revised manuscript file. Where possible, we have also added additional supplementary figures, at the request of the reviewers, or to support our rebuttal statements. 

We believe the reviewers’ comments and the subsequent changes made to this manuscript have significantly improved the quality of the article and we thank them for this contribution.

Please note that the image quality of the submitted figures is diminished when converted to PDF for submission by the PLOS software and does not reflect the image quality of the original files.

---

## [Decision Letter · Decision Letter 1]

10 Jun 2022

PONE-D-21-32716R1Defining Cardiac Cell Populations and Relative Cellular Composition of the Early Fetal Human HeartPLOS ONE

Dear Dr. Dewing,

Thank you for submitting your manuscript to PLOS ONE. After careful consideration, we feel that it has merit but does not fully meet PLOS ONE’s publication criteria as it currently stands. Therefore, we invite you to submit a revised version of the manuscript that addresses the points raised during the review process.

Two reviewers still have significant concerns about the reporting of the methodology and are not satisfied with the authors’ responses mainly because they have made no changes to the interpretation of the data as suggested and, importantly, they have not addressed even some major comments. Further, it is really annoying when authors claim they have introduced several changes to the manuscript but they did not and it is not acceptable that they are not able to present the raw data when requested.  

I decide for a major revision but I would be willing to reconsider a revised submission only once **all the concerns are addressed **and if they are addressed** within and not later than 40 days (mandatory due date: July 21^st^ 2022)**

Please submit your revised manuscript by **July 21^st^ 2022**. Please include the following items when submitting your revised manuscript:A rebuttal letter that responds to each point raised by the academic editor and reviewer(s). You should upload this letter as a separate file labeled 'Response to Reviewers'.A marked-up copy of your manuscript that highlights changes made to the original version. You should upload this as a separate file labeled 'Revised Manuscript with Track Changes'.An unmarked version of your revised paper without tracked changes. You should upload this as a separate file labeled 'Manuscript'.

We look forward to receiving your revised manuscript.

Kind regards,

Federica Limana

Academic Editor

PLOS ONE

Reviewers' comments:

Reviewer's Responses to Questions

**Comments to the Author**

1. If the authors have adequately addressed your comments raised in a previous round of review and you feel that this manuscript is now acceptable for publication, you may indicate that here to bypass the “Comments to the Author” section, enter your conflict of interest statement in the “Confidential to Editor” section, and submit your "Accept" recommendation.

Reviewer #1: (No Response)

Reviewer #2: All comments have been addressed

Reviewer #3: (No Response)

2. Is the manuscript technically sound, and do the data support the conclusions?

Reviewer #1: Partly

Reviewer #2: Yes

Reviewer #3: No

3. Has the statistical analysis been performed appropriately and rigorously? 

Reviewer #1: N/A

Reviewer #2: Yes

Reviewer #3: N/A

4. Have the authors made all data underlying the findings in their manuscript fully available?

Reviewer #1: Yes

Reviewer #2: Yes

Reviewer #3: No

5. Is the manuscript presented in an intelligible fashion and written in standard English?

Reviewer #1: Yes

Reviewer #2: Yes

Reviewer #3: Yes

6. Review Comments to the Author

Reviewer #1: The authors have added further details to the methods section and provided a little more clarity in some figure legends. They have also inserted some additional supplemental data, including negative controls for IHC staining and a table showing the antibodies used. Together these are useful to clarify what was done. In particular, they have explained what the negative control data for the flow analysis represents. This is helpful, strengthens their conclusions and addresses some of my original concerns. I believe this revised version now satisfies PLOS guidelines on minimal reporting.

However, they have made no changes to the interpretation of the data based on the feedback received. The figures, abstract and main text remain largely unchanged. Despite providing a detailed rebuttal to the reviewers, very few changes have been made to the manuscript itself. Most of the points made in the rebuttal are omitted from the main manuscript. For this reason, some of my original concerns remain outstanding.

In general, I feel that the manuscript that the manuscript would benefit from a re-write. It remains thin on experimental details and I feel that some sections could be explained more clearly. The sub-headings could be changed to describe the key findings rather than the methods used. More explanation is needed in the text of the fibroblast explant culture method and also of the rationale of analysing different parts of the heart.

Detailed Comments

Fig1

Negative controls for these antibodies are now shown in Fig S2 and look fine. This figure should be referred to in the text.

Fig2

This figure has not been modified in the revised version.

The authors have now told us what the negative control is and this has answered some of my concerns. However, the positioning of the gate remains arbitrary and therefore all of the concerns I raised in my original review regarding this remain outstanding.

In their rebuttal, the authors refer to a figure from Banerjee 2007, which they state was used to design their own figure. I would therefore like to point out that in Banerjee the position of the gate is not arbitrary, but is aligned with the end of the unstained control cells, and is in the same position in every panel. The position of the gate in this current manuscript is arbitrary. (the authors correctly point out that Banerjee does not use an isotype/secondary control to position the gate. Their method is indeed an improvement, but only if information from this control is used to inform positioning of the gate).

Much of the paper concerns the existence of two populations of cells (high and low expression of a given marker), therefore the decision to arbitrarily exclude cells showing low MHC expression is bizarre.

The positioning of the gate in other panels also appears rather random, sometimes overlapping the tail of the negative peak, sometimes not.

Fig2a: The purpose of this figure is to show autofluorescence of unlabelled cells. However, the x axis is unlabelled so it is unclear which channel this corresponds to, making this panel meaningless. The authors used antibodies with a variety of different fluorophores, therefore, we need to see autofluorescence data plotted for each channel. It would be better to plot these unlabelled cells on each panel along with experimental and control antibodies.

Fig3

No Fig 3 is provided.

Fig 4

OK

Fig 5

It is still not too clear what we are looking at. It would be better if the panels were at the same mag and showed adjacent sections of the same tissue. One concern I have here is that in Fig 5a the line dividing the myocardium from the blood vessel appears to be based on differing expression levels of cTnI, yet this difference is then given as evidence that non-cardiomyocytes express cTnI. This is circular logic, and could be avoided if another image showing MHC/SMA were included and used to draw the line.

Figs 6, 7

Some of the expression patterns shown in Fig 7 are quite odd. cTnI is shown expressed in the nuclei of these fibroblasts – this is very odd for a structural protein and not the expected pattern. The expression pattern shown in Fig S7 is quite different to that in Fig 7. However, neither of these are points are mentioned in the text at all. It looks like something has gone wrong here.

I think Figs 6 and 7 could be combined into one, perhaps with a diagram to explain the methods. Co-expression data shown in Fig S7 is key, so could be moved to main figure.

Reviewer #2: I would like to thank the authors for addressing all of my comments. I have no additional suggestions.

Reviewer #3: The manuscript “Defining Cardiac Cell Populations and Relative Cellular Composition of the Early Fetal Human Heart” tries to analyze commonly used markers for the main cardiac cell types, such as cardiomyocytes, cardiac fibroblasts, endothelial cells and smooth muscle cells, in human fetal hearts (< 12 pcw ). The findings are mainly based on antibodies used for immunohistochemistry as well as flow cytometry analyses.

In the revised version of the manuscript, the authors improved the picture quality for immunohistochemistry. The authors also added more details to the Material & Method’s section. In addition, the authors added a paragraph to the discussion where they compare their findings about the human fetal heart to mouse data and to state-of-the-art single cell data.

However, there are still several concerns from my side.

Unfortunately, the authors did not address some of my major comments. They did not even answer to points such as “In addition, it would be helpful to directly compare the findings to adult specimen. In the discussion, the authors talk about the differences in cell composition and marker expression between fetal hearts and adult hearts. But given the different antibodies used for detection of cell populations and different methods used for dissociation of tissues, such numbers might be highly variable. Please perform immunohistochemical stainings and flow cytometry analyses with the same antibodies on adult specimen.”

The comment “The newest reference cited by the authors is from the year 2019. However, science is rapidly progressing, especially on the topic of cardiac development and fetal hearts since the heart is one of the most important organs and the one developing first in embryogenesis. References should be updated and newest publications, especially concerning single cell analysis of fetal human hearts should be included“ was addressed by adding one more reference (from 2020). However, also here the authors did not answer to my comment.

I see that the authors might feel annoyed if they get lots of comments from the reviewers. However, as a reviewer, one invests time to perform this review and really tries to offer comments that help to improve the manuscript.

At least, the authors addressed some of my specific concerns. However, sometimes they claim that they have changed a certain paragraph or statement but they did not.

In addition, I see it as a significant problem that, in several cases, the authors are not able to present the source or original data or images.

Please find the problematic points in the following.

Specific Concerns:

my comment: p4, line 84: CD31 is not specific for endothelial cells (is also expressed in macrophages), alphaSMA is not specific for Smooth muscle cells but also expressed in activated fibroblasts. Please relativize that.

authors: We have now added: ‘However, CD31 is also present on macrophages, and α-SMA is upregulated in activated fibroblasts, termed ‘myofibroblasts’, that respond to injured cardiac tissue.’

Unfortunately I cannot find that in the text. Might the authors have sent a wrong version of their revised text?

p5: Materials and Methods section:

my comment: p5, line 109: heading should be “Fibroblast Isolation and Cell Culture”

authors: This has now been amended (on page 7, line 152)

it has not been changed. Again: wrong text version?

my comment: p7, line 143: please note in that paragraph that results of the RT-PCR were visualized on an XXX% agarose gel and the fluorescent dye you used….

authors: This has now been added to the end of the RT-PCR section

it has not been added. Wrong text version?

my comment: p7: Results

p7, line 157: TnT is not shown in Fig1A? Missing? Not stained?

authors: We used cTnT to validate the flow cytometry data of cTnI, showing high and low positive populations (Supplementary Figure 3). We did not do any additional IHC with this antibody, although this antibody is well characterised, having been cited in over 71 publications.

Then it should not be mentioned when talking about the IHC results in Figure 1. Please change that.

my comment: p7, line 161: since Thy1 is a surface protein, it should not be localized in the cytoplasm…. Vimentin should mark fibres of the cytoskeleton (However, I cannot recognize these patterns since the picture quality is so poor). In addition, there is DDR2 in the figure (not mentioned in the text).

authors: We have now amended this figure to include images of higher quality that better represent the expression we saw in the heart. We agree that as a surface marker we would expect to see Thy-1 on the cell surface, however, we have identified other published IHC and immunocytochemistry that show Thy-1 expression not localised only to the cell surface (Sidney et al., 2015, DOI: https://doi.org/10.1016/j.jcyt.2015.08.003.; Pietro et al., 2020 DOI:https://doi.org/10.3390/ijms21124356). This suggests possible internalisation of the receptor where it may play an alternative role. We must also consider that the thickness and cellular density of the tissue, which will result insectioning of cells along different planes and therefore surface staining can be interpreted as cytoplasmic based on localisation relative to the nucleus.

I agree with the author’s remarks. However, the text says Thy1 is “predominantly” found in cytoplasm (I would not agree with that from what I see on the pictures, now that the picture quality is improved). You might write: is “also” found in cytoplasm….

authors: We have now also added in commentary of DDR2 expression pattern in the text.

No, I cannot find that.

my comment: p8, line 181/182 (legend for Fig 2I): Percentages of MHC and TnI are not the same as mentioned in the text???? Please correct.

authors: We have checked that the percentage of MHC and TnI in the text are the same as those displayed in the figure.

But the bar graph in Fig 2I shows not the correct percentages for MHC (the text says 75%, the bar graph says 65%) and TnI.(the text says 93%, the bar graph says something less than 40%). Please correct that.

my comment: p8, line 187/188: Please indicate percentages for TnT high and low expressing populations. By the way, the same holds true for MHC. Can you also add this into the Suppl Figure?

authors: Unfortunately, we are not able to source this data.

then the data should not be used for a manuscript.

my comment: In Fig 6 you should show a negative control and an amplification control such as GAPDH or BetaActin for all the lines so that one can decide if the differences in the band intensities are really reflecting differences in gene expression and not only differences in amplification efficiency, etc.

authors: Housekeeping controls were used for reverse transcription PCR (RT-PCR), however, unfortunately the primary data (gel images) are no longer able to be accessed. We used RT-PCR to show the presence or absence of transcripts and to confirm there was no contamination of CMs in the fibroblast population. We have now amended the text to ensure we only discuss the RT-PCR data in a qualitative manner.

data used in a manuscript should be available. Apart from this, it is only a RT-PCR, so it could easily be repeated with a housekeeping gene.

Figures

my comment: The Figure Legends should also be improved. In the legends you should shortly describe what you see in the pictures, so that you can understand without reading the text.

authors: We have amended the figure legends where necessary.

Actually the Figure legends were not changed

authors: We believe these legends are consistent with those from previous PlosOne publications. If the editor would like us to amend further we are happy to do so.

7. PLOS authors have the option to publish the peer review history of their article (what does this mean?). If published, this will include your full peer review and any attached files.

Reviewer #1: **Yes: **Iain Dykes

Reviewer #2: No

Reviewer #3: No

---

## [Author Response · Author response to Decision Letter 1]

19 Jul 2022

For the editor: please see attached 'cover letter' file that has been uploaded

For the reviewers: Please see attached 'response to reviewers' file that has been uploaded

---

## [Decision Letter · Decision Letter 2]

31 Aug 2022

PONE-D-21-32716R2Defining Cardiac Cell Populations and Relative Cellular Composition of the Early Fetal Human HeartPLOS ONE

Dear Dr. Dewing,

Thank you for submitting your manuscript to PLOS ONE. After careful consideration, we feel that it has merit but does not fully meet PLOS ONE’s publication criteria as it currently stands. Therefore, we invite you to submit a revised version of the manuscript that addresses the points raised during the review process.

The manuscript has been seen by three reviewers and they have raised a number of suggestions. Specifically they feel that the explanations you have provided in the rebuttal letter are very good and should be included in the text to provide more clarity. They also raise concerns about the description of some of the methods that could be improved.Could you please revise your manuscript to include their recommendations?

We look forward to receiving your revised manuscript.

Kind regards,

Thomas Tischer

Staff Editor

PLOS ONE

Journal Requirements:

Reviewers' comments:

Reviewer's Responses to Questions

**Comments to the Author**

1. If the authors have adequately addressed your comments raised in a previous round of review and you feel that this manuscript is now acceptable for publication, you may indicate that here to bypass the “Comments to the Author” section, enter your conflict of interest statement in the “Confidential to Editor” section, and submit your "Accept" recommendation.

Reviewer #1: (No Response)

Reviewer #2: All comments have been addressed

Reviewer #3: All comments have been addressed

2. Is the manuscript technically sound, and do the data support the conclusions?

Reviewer #1: Yes

Reviewer #2: Yes

Reviewer #3: Yes

3. Has the statistical analysis been performed appropriately and rigorously? 

Reviewer #1: N/A

Reviewer #2: Yes

Reviewer #3: N/A

4. Have the authors made all data underlying the findings in their manuscript fully available?

Reviewer #1: Yes

Reviewer #2: Yes

Reviewer #3: Yes

5. Is the manuscript presented in an intelligible fashion and written in standard English?

Reviewer #1: Yes

Reviewer #2: Yes

Reviewer #3: Yes

6. Review Comments to the Author

Reviewer #1: In the latest revision (R2), the authors have corrected some errors that I highlighted in the R1 version, included further control data in the supplement, improved some figures and provided a detailed explanation of their methods in the rebuttal. However, what they have not done is make any major changes to the text of the manuscript. The revised R2 manuscript actually differs very little from R1 (the tracked changes version provided shows changes made in R1, it is difficult to find many new changes for R2). It is clear that the authors are very much unwilling to make significant changes to the manuscript, despite extensive comments/concerns by myself and Reviewer 3. As far as I can gather from the tracked changes shown, very few changes have been made to the text of the manuscript: no more that 2 or 3 sentences have been added.

While I appreciate that the authors are perfectly entitled to have a different opinion on how the data should be analysed and interpreted, I think it is important that they include enough detail in the text to allow a reader to clearly follow what was done. I’m afraid I still don’t believe that this has been achieved. I find the manuscript quite hard to read because the authors provide minimal information in the text to explain what they have done, and why. The detailed explanation provided in the rebuttal is not included in the text of the manuscript. There are also no figure legends. Last time I suggested that a full re-write of the manuscript was needed, and I therefore find it rather frustrating that the authors have made only minor changes. As stated previously, I feel that many of the points made in the rebuttal should be incorporated into the manuscript itself to make it easier to follow, this has simply not been done.

To address my previous comments:

"However, they have made no changes to the interpretation of the data based on the feedback received. The figures, abstract and main text remain largely unchanged. Despite providing a detailed rebuttal to the reviewers, very few changes have been made to the manuscript itself. Most of the points made in the rebuttal are omitted from the main manuscript. "

This is still the case, very little has changed since R1.I do not think the conclusions of the paper have been changed at all despite this extensive review process.

"In general, I feel that the manuscript that the manuscript would benefit from a re-write. "

Not done, only minimal changes to text.

"It remains thin on experimental details and I feel that some sections could be explained more clearly. The sub-headings could be changed to describe the key findings rather than the methods used."

The authors rejected this suggestion in the rebuttal.

"More explanation is needed in the text of the fibroblast explant culture method and also of the rationale of analysing different parts of the heart."

The authors state in the rebuttal that these have been added, but I was unable to find much in the manuscript. A single line has been added to the methods. The results section still contains almost no explanation of the rationale behind the explant experiment, this is simply not sufficient to follow.

Fig2

The authors have answered my concerns in the rebuttal, but I feel these details need to be explained fully in the text of the manuscript itself.

Furthermore, if the dot blots were used to set the gates, then why not put these images in the main figure rather than in the supplement?

Fig 5

Again, my concerns have been answered in the rebuttal, but I feel that this explanation needs to be in the text of the manuscript.

Figs 6, 7

"Some of the expression patterns shown in Fig 7 are quite odd. cTnI is shown expressed in the nuclei of these fibroblasts – this is very odd for a structural protein and not the expected pattern. The expression pattern shown in Fig S7 is quite different to that in Fig 7. However, neither of these are points are mentioned in the text at all. It looks like something has gone wrong here."

The authors have corrected this error

"I think Figs 6 and 7 could be combined into one, perhaps with a diagram to explain the methods. Co-expression data shown in Fig S7 is key, so could be moved to main figure."

The authors have made a combined figure, but have not included a diagram to explain the methods. As stated a few times now, I feel this experiment is unclear.

Reviewer #2: (No Response)

Reviewer #3: My concerns have been addressed now.

Only one little thing: please correct at p10, line 217: not cTNI but cTNT

7. PLOS authors have the option to publish the peer review history of their article (what does this mean?). If published, this will include your full peer review and any attached files.

Reviewer #1: **Yes: **iain dykes

Reviewer #2: **Yes: **Anis Hanna, MD

Reviewer #3: No

---

## [Author Response · Author response to Decision Letter 2]

21 Oct 2022

Please see attached Cover Letter and Response to Reviewer files

---

## [Decision Letter · Decision Letter 3]

8 Nov 2022

Defining Cardiac Cell Populations and Relative Cellular Composition of the Early Fetal Human Heart

PONE-D-21-32716R3

Dear Dr. Dewing,

We’re pleased to inform you that your manuscript has been judged scientifically suitable for publication and will be formally accepted for publication once it meets all outstanding technical requirements.

Kind regards,

Koh Ono, M.D., Ph.D.

Academic Editor

PLOS ONE

Additional Editor Comments (optional):

The manuscript was improved. I have no further comments.

Reviewers' comments:

Reviewer's Responses to Questions

**Comments to the Author**

1. If the authors have adequately addressed your comments raised in a previous round of review and you feel that this manuscript is now acceptable for publication, you may indicate that here to bypass the “Comments to the Author” section, enter your conflict of interest statement in the “Confidential to Editor” section, and submit your "Accept" recommendation.

Reviewer #1: All comments have been addressed

Reviewer #2: All comments have been addressed

Reviewer #3: All comments have been addressed

2. Is the manuscript technically sound, and do the data support the conclusions?

Reviewer #1: (No Response)

Reviewer #2: Yes

Reviewer #3: Yes

3. Has the statistical analysis been performed appropriately and rigorously? 

Reviewer #1: (No Response)

Reviewer #2: Yes

Reviewer #3: N/A

4. Have the authors made all data underlying the findings in their manuscript fully available?

Reviewer #1: (No Response)

Reviewer #2: Yes

Reviewer #3: Yes

5. Is the manuscript presented in an intelligible fashion and written in standard English?

Reviewer #1: (No Response)

Reviewer #2: Yes

Reviewer #3: Yes

6. Review Comments to the Author

Reviewer #1: (No Response)

Reviewer #2: (No Response)

Reviewer #3: See answer above.

All of my comments have been addressed after Revision R2.

I think the authors significantly improved their manuscript by several rounds of revisions.

7. PLOS authors have the option to publish the peer review history of their article (what does this mean?). If published, this will include your full peer review and any attached files.

Reviewer #1: **Yes: **Iain Dykes

Reviewer #2: **Yes: **Anis Hanna

Reviewer #3: No

---

## [Editor Report · Acceptance letter]

18 Nov 2022

PONE-D-21-32716R3 

Defining Cardiac Cell Populations and Relative Cellular Composition of the Early Fetal Human Heart 

Dear Dr. Dewing:

I'm pleased to inform you that your manuscript has been deemed suitable for publication in PLOS ONE. Congratulations! Your manuscript is now with our production department. 

Kind regards, 

on behalf of

Dr. Koh Ono 

Academic Editor

PLOS ONE